# Caring for Daughters with Anorexia Nervosa: A Qualitative Study on Parents’ Representation of the Problem and Management of the Disorder

**DOI:** 10.3390/healthcare10071353

**Published:** 2022-07-21

**Authors:** Luna Carpinelli, Tiziana Marinaci, Giulia Savarese

**Affiliations:** Department of Medicine, Surgery and Dentistry “Scuola Medica Salernitana”, University of Salerno, 84081 Baronissi, Italy; lcarpinelli@unisa.it (L.C.); tizianamarinaci@gmail.com (T.M.)

**Keywords:** anorexia nervosa, eating disorders, parents, primary caregivers, disorder management, dimensions of meanings, qualitative study

## Abstract

Background: This study explores the implicit theories by which primary caregivers (PC) of patients diagnosed with anorexia nervosa (AN) understand the eating disorder and interpret their role in treating and managing the problem. Methodology: In-depth, semi-structured, and open interview questions were used to achieve the study’s goals. In total, 19 caregivers, 16 mothers, and three fathers (mean age: 50.74; SD: 5.248) from a public service for the treatment of behavioral disorders in southern Italy were interviewed. A lexical correspondence analysis (LCA) was applied to the verbatim transcripts to identify the main factorial dimensions, which organize similarity and dissimilarity in the collected discourses. Results: The first dimension represents the dialectic between two different models of explanation of the problem, and the second dimension represents the dialectic between two different perspectives on the attribution of responsibility. Overall, the analyses show the difficulties of PC in exploring the emotional dynamics of the problem and the tendency to take out of the family context every possible representation of the role that it can play in the maintenance and evolution of the disorder. Conclusions: The strategies to prevent and treat AN may benefit from knowledge of the meaning’s lenses adopted by the primary caregivers to explain and cope with their daughters’ illness.

## 1. Introduction

Within the cluster of eating disorders (EDs), anorexia nervosa (AN) is a psychiatric disorder that has a negative impact on both an emotional and physical level [1]. At the same time, it is a pathology that not only affects the person suffering from the disorder but also the entire family context [2,3]. In fact, the symptoms manifested put a severe test on the relational skills of family members, often causing enormous changes in relationships and daily routines.

In January 2022, the Italian National Institute of Health showed data from the first census regarding the prevalence of AN in Italy [4]. This report identified that 90% of those with AN are female and 10% are male. The age distribution for the diagnosis is 59% (age 13–25 years) and 38.6% < 12. Specifically, the age of onset for males is between 17–24 years while for females it is between 15–18 years.

Additionally, the COVID-19 pandemic has negatively affected mental health problems; in particular, there was a 30% increase in cases of AN compared to 2019 data [5].

Generally, parents of children with an EDs member report worse family functioning and specific problems with communication, flexibility, and cohesion [6,7]. In a study [8] in which the opinions of parents and siblings of AN patients (n° 34 female patients; mean age 15.7) were evaluated, it emerged that parents rated their families with a high level of conflict and with a more dysfunctional general functioning especially with regard to the relationship with the mother. They experience the illness as one of great suffering and stress, feel that they are dominated by the rules imposed by AN, and perceive emotions of intense tension and frustration. They perceive the feeling of losing control and are very afraid of their child’s prognosis [9], particularly after diagnosis.

The disease takes over the entire life of parents, especially that of mothers, who wake up and go to sleep with the same thought: monitoring food and exercise. In most cases, a parent’s life is organized around AN so as not to leave their child alone and to ensure that the situation does not worsen. All of this leads to the neglect of one’s own needs and requirements since the goal is not one’s own well-being but rather that of one’s children [10].

Currently, scientific research, through multiple transversal and longitudinal studies aimed at evaluating the role of family interactions in the etiopathogenesis of AN, has attested both to the existence of a multiplicity of different relational patterns in the families of patients with AN and to the presence of a complex intertwining of different factors underlying the etiology of this disorder.

Caregivers, according to Coomber and King [11], reduce the use of adaptive coping strategies and implement moderate levels of maladaptive coping strategies, such as self-blame, denial, and behavioral disengagement with their cared-for child. Weaver [12] tried to explain the reasons for this disengagement, concluding that parents, horrified by the EDs, try to disengage from associated behaviors (e.g., outbursts of anger, binges, and covert actions), out of fear of making matters worse, for their need to take a thoughtful step back to plan their next action, and for lack of practical knowledge.

In general, parents of patients with AN use coping skills mainly focused on the problem because they are necessary to continue the work of the constant supervision of the sick child [13] and also to avoid the burden related to the treatment. This burden includes both an objective and subjective form. The objective burden is characterized by finding time for clinic appointments, assistance in daily living activities, and financial support. The subjective burden includes distress, emotional tension, difficulties in one’s social life and level of dissatisfaction. Hillege et al. [14] attributed the causes of the AN patient’s caregiver burden to parents’ perceptions of strangeness and reported “caregiver isolation” from friends, relatives, and social networks. Parents also reported experiencing a lot of pain, embarrassment, and fear as well as being ostracized by healthcare workers who view that the family made the patient sicker.

Parks et al. [15] conducted a comparative study to assess caregiving experience and coping strategies among groups of parents caring for their children with different diagnoses. The study sample consisted of 48 mothers (mean age 44.88) and 44 fathers (mean age 47.45) of 50 patients diagnosed with EDs (98% daughters; mean age 14.78); 46 mothers (mean age 49.95) and 36 fathers (mean age 51.06) of 46 patients diagnosed with substance abuse (82.6% sons; mean age 18.3) and 63 mothers (mean age 47.56) and 50 fathers (mean age 49.72) of 66 healthy adolescents (all daughters; mean age 14.39). Analysis of the results showed that mothers of adolescents with EDs adopt self-sustaining strategies that focus on problems more than mothers of healthy adolescents, and this can lead to hostility and over-involvement. Some differences in parental reactions between mothers and fathers have been found, the former being more emotionally involved, less critical and more prone to seeking supportive social relationships than fathers.

Differences between mothers and fathers are also highlighted between negative emotions and a sense of self-efficacy: mothers often manifest fear, and this produces accommodating attitudes towards the disease of their daughter AN and predicts their sense of self-efficacy; among fathers, neither fear nor self-efficacy predicts disease [16].

This study aims to fill this gap in the literature in patients with AN by exploring the differences in coping and disease-management strategies between mothers and fathers which have not yet been sufficiently evaluated [15], as well as the perception that parents understood as primary caregivers have their own children’s illness. Given the lack of in-depth previous studies, we had no a priori hypotheses about the specific associations, differences, and characteristics that emerged.

## 2. Theoretical Framework

According to Polkinghorne [17], narrative is the “linguistic form uniquely suited for displaying human existence as situated action. Narrative descriptions exhibit human activity as purposeful engagement in the world” (p. 5). Frank [18] notes that while people tell their own stories of disease, they compose these stories by adopting and combining the narrative types that cultures make available to them. In this sense, narrative telling’s are not mere conversational realities, but themselves represent constituents of ongoing and often institutionalized patterns of social conduct [19]. The narratives of health and illness, both personal and social, are how people make sense of their experiences [20] and are part of general social relationships and cultural values that characterize a specific life context [21].

Narratives are how people make sense of their experiences of illness [20]. From a cultural and social constructive perspective [22,23,24,25,26], the ways in which people experience a disease can be understood as an intimate part of the social system of meaning [27,28]. In this paper, the lens of semiotic cultural psychological theory (SCPT) [29,30,31] is adopted. The SPCT, positioned within the socio- constructionist area [32,33,34], assumes a vision of the individual as a semiotic subject (or creator of meaning) [35,36], i.e., a subject continuously engaged in the interpretation of experience [29,37,38]. In this sense, meaning is not attributed to the contents of the world that exist before being interpreted. Rather, sensemaking makes up the reality; namely, the way of presentifying its reality to the mind [20,31]. Therefore, meaning constructs the actual content of the experience of both the external and internal environment (i.e., the experience of the body and feelings of each one), and more generally, the image that individuals have of themselves and their relationships within the context, i.e., their social identity. Indeed, the ways of representing reality guide decisions and approach strategies to problems, enable attitudes, orient behaviors and, in this way, build worlds [39]. Several studies have highlighted how interpretations are a way to be channeled to act and react in a certain way in different areas of life: for example, they orient the methods of evaluating a public service [40], the ways of relating with otherness [41], people’s attitudes towards care, and intervention systems [42]. Within AN, anthropological studies have highlighted how cultural values and practices can inform specialist treatment settings for EDs and can inform treatment dynamics and potentially cure rates. Hospital spaces themselves, such as patient bedrooms in inpatient units, can potentially represent a replica of patient bedrooms at home, which are richly associated with the rituals and private cognitions that perpetuate AN [43,44]. At the same time, the context, as well as social and cultural dimensions, can intervene in influencing the same theories on AN, the models of intervention and treatment of the disorder [45]. For example, anthropological studies show that US physicians are prone to conceptualize eating disorders as being caused by thwarted individuation, while in Mexico, treatment outcomes are positively assessed as women become socially integrated and receptive to their communities [43,46]. The representations and the way of narrating the implications of the disorder, the conditions of care and taking charge define and shape their specifically relationship with the services and the responsibility with respect to the onset, development and maintenance of the problem.

In this perspective that the need arises to shift the gaze from the disturbance to the way of representing it, from the fact to the narration that emerges from the experiences of people. That is acknowledging the close interconnection of stories with the subjective and intersubjective world of people.

This hinges on the general idea that it is possible to suspend the belief that the categories proposed to describe problems and solutions receive their legitimacy from the observation and examination of “reality”. It is to engage in a careful understanding of the context (subjective, intersubjective, cultural) which gives meaning to a certain way of feeling and acting, even when this sense escapes the scrutiny of logical–analytical thought [44].

Consequently, the importance of adopting the primary caregiver perspective is fundamental in order to respond to the specificity of their ask for help. That is, acknowledging that valuing their narratives of disease representation and management as “data that matters” lies in recognizing their impact on the ways in which they experience, evaluate, face and react to the planned actions and to the standardized care of the services.

## 3. Materials and Methods

### 3.1. Participants

The participants in this study were recruited from patients treated consecutively in a public outpatient clinic specializing in EDs in the Campania region (Southern Italy). All came from different areas of the same region. The care service is a regional reference center. All patients (n° 13 female; mean age = 16.92; SD = 3.73) are from a previous hospital stay and have a diagnosis of AN, in accordance with DSM 5 criteria [47]. For each of them, the two primary caregivers actively involved in patient care were invited to participate in the study. Of the 12 couples involved (a widowed father and a divorced mother unaccompanied by her daughter’s father), in only two cases did the fathers decide to participate in the research. The remaining ten fathers agreed to participate but delegated it to the mother.

Demographic characteristics (age, marital/living status, and occupation) were gathered during the interview. Disease duration was assessed by considering the time elapsed between the onset of symptoms and the time of hospital treatment. A total of 60% of family members declared a course of care between one and two years or more, 30% for less than a year, and 10% for no more than six months.

The final sample of the study (see Table 1) included 19 parents (mean age = 50.74; SD = 5.24), of which 84.2% were mothers and 15.8% were fathers. Table 1 shows the socio-demographic characteristics of the sample. Twenty-six percent were divorced and 5.3% were widowers. As regards the qualifications obtained, 5.3% had a lower primary school diploma, 26.3% had a lower secondary school education, 36.8% had a high school diploma, and the remaining 31.6% had a university degree.

Regarding the variable “employment,” the sample is composed as follows: 31.6% were employed in a public or private company, 15.8% were housewives, 31.6% were freelance professionals, 15.8% were unemployed, and 5.3% were retired.

All procedures performed in the study complied with the ethical standards of the institutional research committee and with the 1964 Declaration of Helsinki and its later amendments or comparable ethical standards. Participants were informed about the general aim of the research, the anonymity of responses, and the voluntary nature of participation and signed informed consent. No incentive was given for participating in the study.

### 3.2. Instrument

In-depth, semi-structured, and open interview questions were used to explore primary caregivers’ implicit theories about the problem of anorexia nervosa and how they make sense of their care role. Each subject was asked to relate their experiences of the disorder, on taking charge of the problem, and on the constraints and resources encountered in the treatment process (Appendix A). The question building process involved a first phase of literature review and subsequent brainstorming among researchers. Each parent voluntarily participated in the interview phase, which was agreed and carried out individually with each parent. The average interview time was 30 to 40 min.

## 4. Data Analysis

The whole corpus of narratives was analyzed through an automatic procedure for content analysis (ACASM) [48,49,50], performed by T-Lab Plus 2022 software [51]. Through this procedure, it was possible to map the main dimensions of meanings underlying the content set. The method is based on the general assumption that meanings are formed in terms of lexical variability. Accordingly, the method aims to detect the ways in which words combine (i.e., co-occur) within utterances, regardless of sentence referentiality [52].

Compared to most semantic analysis methods focusing on the co-occurrence of lexical units, the main specificity of ACASM is that it adopts a single sentence or a group of a few sentences as the unit of context (the unit of context is the segment of text within which co-occurrences are detected). The analysis is designed to map the variability of meaning—the set of different themes (e.g., beliefs, opinions, combined statements, and verbal images) that emerge in response to the open stimulus—structurally. Each dimension can be conceived as a component/quality of the topic that has been made relevant by the participants and offers space for a plurality of statements and positions. The structural map of the variability of meaning therefore goes beyond the descriptive level of the content analysis and identifies the semantic structure that generates the variability of the contents [48,49,53].

The ACASM procedure consists of the following steps:

First, through an automated procedure [54], the textual corpus of the narratives is segmented into elementary context units (ECUs). Second, lexical forms (i.e., any string of characters comprised between two blank spaces) present in the ECUs are subject to a lemmatization procedure (i.e., lexical forms “son,” “sons” and “daughter” are classified as the lemma “child”). Lexical forms devoid of meaning (e.g., or, so, because, “that is” etc.) are automatically excluded. A digital matrix of the text is generated, having as rows the ECU and the lemmas as columns. The ij-th cell assumes a value of 1 if the j-th lemma is contained in the i-th ECU; otherwise, a value of 0 is assigned.

Lexical correspondence analysis (LCA) is a factor analysis procedure that works on nominal data [55]. In general terms, the method breaks down the whole lexical variability (i.e., the distribution of the lemmas presented in the corpus) in terms of a multidimensional structure of opposed factorial polarities. Each factorial dimension describes the juxtaposition of two strongly associated (co-occurring) lemma patterns and can be interpreted as a marker of a latent dimension of meanings underlying (dis)similarities in respondents’ discourses [48]. The interpretation of the factorial dimensions is performed in terms of the inferential reconstruction of the overall meaning predicted by the set of co-occurring lemmas associated with each polarity. In this approach, the semantic content of individual words is interpreted in terms of co-occurrences in order to capture the underlying global meaning. The first two factors extracted from the LCA were selected as those explaining most of the inertia in the data matrix. In a double-blind procedure, the authors, two development and education psychologists and one clinical psychologist, all with previous experience in qualitative research, defined the labels and commented on the results. After reading the analysis as a whole, all authors performed a second in-depth reading of the corpus of interviews, and discussed and compared the results until an agreement was reached.

Finally, with the support of the same software, a specificity analysis (SA) was applied to the entire corpus of text to identify and investigate other differences or similarities in the respondent’s discourses. SA is a comparative technique to identify words that are typical and statistically over- or under-used or even exclusive in a segment of the corpus, based on a chi-square statistic.

Table 2 describes the characteristics of the dataset.

## 5. Results

Table 3 and Table 4 illustrate the first and second factorial dimensions obtained by the ACASM procedure, respectively. For each polarity of the two dimensions, the lemmas with the highest level of association (V-Test) are reported, as well as their interpretation in terms of the labeling of their meaning. Henceforth, we adopt capital letters for labeling the dimensions of meaning and italics for the interpretation of polarities.

### Dimensions of Meanings

First Factorial Dimension. Models of Problem Explanation: onset versus (no explanation)

This dimension opposes two patterns of lemma, which we interpret as the markers of two different models of problem explanation (Table 3).

*Onset* (-): On the negative polarity, the discourses focus on the onset (origin) of the problem, connected to aspects perceived as missing or problematic (lack, problematic, pathology, prevention), both at the family (family, affection, prevention) and personal level (self-esteem, need) (*then everyone expresses their refusal based on the problem that can be familiar or personal*).

In the discourses of family members, anorexia seems to emerge as a product of an individual and/or family deficiency that has not been remedied in time.


*“In my opinion, what starts immediately is the lack of self-esteem. It is also a question of character, perhaps, which then leads to being more affected by this pathology I think.”*

*(Mother, 49 years old)*



*“I think it was the death of a wife, missing her, I think so. Or maybe in high school, some friends, unfortunately this also happens. First she was fine, then someone out of envy put in her brain that she had to lose weight, now she is eating again, maybe she has passed the step.”*

*(Father, 64 years old)*



*In any case, the onset of the disease seems to be an interlock, generated by the family or by the patient’s innate pathology, from which it seems almost impossible to escape once the disease has manifested itself: “I repeat it can arise from a family context with its problems or it can be genetic and also accompanied by a personality disorder, in the sense I do not accept myself, I do not like myself and I have to change, to change I have to do this.”*

*(Father, 49 years old)*



*“Because it is a monster that takes over, a very ugly feeling, of helplessness more than anything else.”*

*(Mother, 48 years old)*


Even the discourse on prevention seems polarized on social or cultural gaps. Anorexia is represented as something that happens suddenly, is challenging to contain before its manifestation, or is a response to society’s shortcomings, guilty of promoting exaggerated dictates of beauty and not providing the necessary information.


*“I don’t know how to prevent it because maybe I found myself inside, catapulted out of nowhere.”*

*(Mother, 49 years old)*



*“[…] So more information, less perfect projected images. So more helpful information to prevent this should be the foundation of our society.”*

*(Mother, 48 years old)*


POLE (−)

*(no explanation)* (+): On the positive polarity, no lemma characterizes the factor. On the part of the parents, there seems to be an absence of representability of possible alternatives with respect to the explanation of the problem. This is congruent with many statements also found in our interviews (*“**I can’t answer, I don’t know if I answered correctly… I didn’t realize the problem I was facing”, “**I didn’t even know what it was”* and *“I really didn’t know what to do… who to contact… and what was best to help her, to solve the problem”*) or with the silence that in some cases accompanied the answers of some of the parents interviewed.

Second Factorial Dimension. Attribution of Responsibility: healthcare services versus society.

This second dimension opposes two lemma models that we interpret as indicators of two ways of interpreting the attribution of responsibility (Table 4).

*Healthcare services* (-): on the negative polarity, the privileged interlocutor of the parents’ speeches are the care services (Polyclinic, ASL, Salerno, Antimo, structure, center), the specialists who deal with them (doctor, psychologist, physician/private psychologist) and the mandate assigned to them (hospitalize, accept, send, call, address, know, resolve). Here, doctors seem to be the only custodians of the fate of their children, called upon to accept the patients and solve the problem (*without the help of a specialized structure, the problem cannot be solved*).


*“[…] that is, after the nutritionist had pointed me to the hospital, before the polyclinic and the hospitalization, we did the day hospital with another nutritionist and then we went downstairs with the doctor. Just that, we didn’t go straight to the hospital I mean. At first once a week to do the day hospital with the nutritionist who weighed her and gave her a diet and then later the girl did not follow everything. Then the doctor sent us to the doctor and then the doctor said we have to hospitalize her, because the girl doesn’t want to participate. And so we found ourselves doing this hospitalization.”*

*(Mother, 45 years old)*


The first step in recognizing the problem for the parents who intervened in our interviews seems to be strictly connected to the search for the right professionalism (doctors, psychologists, nutritionists, etc.) who may be able to take on the whole complexity of the disorder. Eating disorders, such as diabetes, appears to be a disease that can be solved by identifying the right therapy.

The responsibility seems to be completely borne by the doctor or the structure from this moment on, and they are called on to solve the problem, of which only the child seems to be the bearer.


*“When my husband and I understood that there was a problem… and… we immediately thought of going to a psychologist, so we moved immediately, almost immediately… The speech is that the psychologist is very good, available, for heaven’s sake… however, it was not her field… when she diagnosed me with eating disorder, she said that we must go elsewhere, but the doctor herself did not know where to send me, that it was a serious problem… so then I had to look… Let’s say that if there was… if there were structures, let’s say apart from the hospital, the polyclinic and then there is also the structure in Salerno, there are various structures in Italy, but let’s say that it’s not within everyone’s reach, in the sense that it is a problem enter it… you have to weigh 38/37 kg… you have to stand there and then almost risking your life… this I say because my experience of let’s say 3 months in the hospital made me understand exactly this… so it would take more structures, more information channels, so that a pers or how I felt on the high seas… she should feel less alone in that sense….”*

*(Mother, 49 years old)*


*Society (+):* On the positive polarity, the discourses focus on the role of society (world, today, society), understood as the main factor responsible for the illness of one’s children. Society makes the body and the brain sick (food, hunger, psychological, brain, body), it delivers to the wrong models and feeds on the expectations and fears of adolescents (going out, staying, afraid, succeeding, thinking, becoming).


*“*
*However, society imposes on us characters, figures to imitate, which then everything emerged from the acquisition of the mobile phone, because before my daughter has never given problems so they are components that then emerge in this sense in my opinion. Then I am not a doctor, but from what I have been able to understand. I also lived in the hospital. Today’s world is based on images, on perfection, therefore giving the wrong tips and this certainly affects a lot.”*

*(Mother, 49 years old)*



*“Because generally the parent, especially the father, especially in an era like today thinks of giving everything by giving what the children want rather than the dress, the trip, the mobile phone, the money to go out at night, but I mean. Then clearly you discover that perhaps one does not become a parent, one does not do this anymore, and that all this is not enough and then it comes back to what I said before. To a world, to a type of society that goes at two hundred kilometers per hour towards material goods, leaving out everything else and therefore these are all things that can be traced back, let’s say, to the bed of a family that is no longer there in fact and above all these guys they lose self-esteem because they are disoriented.”*

*(Father, 55 years old)*


The narratives of these parents seem to speak of something much greater, something that is difficult to manage, in front of which despite all the *love*, one is *afraid*, one feels helpless and unprepared.

Anorexia is frightening, described as the symbiosis of a mind and a body that cannot react to the countless inputs that the outside world imposes on them. A world that traps and continually puts their existence at risk.


*“It is the refusal of food, in short, it is all a psychological factor. I’s like the brain refuses to eat, this is it. Heart and brain are in symbiosis and everything starts with the brain. […] Another thing could be television, which bombard the brains of models who do not know if they are human beings or walking ibex. So a weak girl can fall into this trap.”*

*(Father, 64 years old)*



*“The fear of losing my daughter. This is little but sure. The fear of losing it, of not being able to do otherwise, because in the face of this situation I did not have the right tools, the right weapons to fight.”*

*(Mother, 49 years old)*


In order to explore whether parental experiences were significantly different according to role and marital status variables, a statistical analysis of specificity was used. It allows us to check the lexical units that are typical and more frequent in a text or a corpus subset defined by a categorical variable. The typical lexical units, defined for overusing (SPEC +) or underusing (SPEC −), are detected by means of the chi-square or the test value computation (see Table 5).

As for the category of mothers, the most characteristic term is the lemma “Doctor.”

Consistent with the interviews, the mothers show themselves as totally absorbed by the practical aspects of care, which include the search for the most appropriate structure where they can treat their daughter in contact with the professional capable of taking care of the disease.


*“The center, that is the private doctor that I met through a family member, because I had no idea who to contact… I informed Dr. Amato, the psychologist. Doctor Manna, don’t worry, don’t regret, let’s see something else, she introduced me to another doctor, I also did an interview with her and then through this doctor, who has a childhood at the polyclinic, I met the center.”*

*(Mother, 55 years old)*


Although these data should be treated with extreme caution, given the small number of respondents, on the other hand, in the discourses of the fathers, terms such as *living, illness, feeling, social, experience, family, keeping*, and *sense* prevail.

The stories of the fathers we met in our interviews tell of a greedy social context that makes them sick and that prevents their daughters from having positive experiences on the world around them.


*“They call them Social but in fact they only create an a-Social, in the sense that everyone lives in a world that burns the moment and lives isolated from the context. I think they are all factors like a little, like drugs, like who comes there out of boredom.”*

*(Father, 54 years old)*


In confirmation of what emerged above, the specificity of the terms (see Table 6) also shows the investment of mothers with respect to the care path of their daughters. Terms such as *following, succeeding, initiating, accepting, facing, losing,* and *ugly, difficult,* and *need* emerge, which seem to once again support the enormous emotional difficulty these women go through in taking charge of their child’s illness, a load that seems to weigh heavily on one’s shoulders (personal, personal) both from an emotional point of view and, as we have emphasized, practical (*nutritionist, Polyclinic*), in the right choice to make with respect to the intervention on the disorder.


*“The fact that, however, I realize that I am not prepared to face such a thing, and then obviously the desire to make my daughter feel good, both psychologically and physically.”*

*(Mother, 42 years old)*


The specificities of the fathers’ concern terms such as *brain, depression, depressive,* and *isolated* show a vision of the problem more oriented to the individual and mental characteristics of the daughter. The responsibility is delegated to the other, “out of oneself”, called from time to time to care (*wife, insiders, you, remit, analysis*), or, in the case of the external context, to the responsibility of the onset and/or maintenance of the disorder (*isolated, quarantine, contributing, disorientation*).


*“No doctor, I think, I would never have noticed. I have to thank my wife and the sensitivity and obviously the natural sensitivity that a mother has towards her children. I only saw that my daughter was losing weight but I certainly did not even have a relationship, let’s say that suddenly, even in terms of dialogue, it was closed. […] Because generally the parent, especially the father, especially in an age like today thinks of giving everything by giving what the children want rather than the dress, the trip, the cell phone, the money to go out at night, but I mean. Then clearly you discover that perhaps one does not act as a parent, one does not act like this anymore….”*

*(Father, 54 years old)*


## 6. Discussion

The aim of our study was to explore the dimensions of meaning (DS) through which primary caregivers represent the disorder of anorexia, and their role in managing the problem and managing care.

The analysis of the narratives based on the ACASM procedure has led to the identification of two main dimensions of meaning that bring two fundamental questions to the fore: how the problem is explained: *onset versus (no explanation*) and how responsibility is attributed *(health**care services versus society*).

Specifically, the first DS (models of problem explanation), the polarity relative to the onset highlights the origin of the problem, is understood as a triggering event capable of generating the disturbance by itself. In the narratives of parents, anorexia seems to be something that suddenly creeps in, takes possession of their children’s lives, and interrupts their “normal perceived trajectory” [56]. Anorexia, when it arrives, like a monster, is able to suddenly burst into family life, simultaneously generating a break with one’s biography, both individual and family.

Consistent with the studies of medical and cultural anthropology, the parent’ narratives are expressed as a means of transmitting the biographical interruption generated by the disease, in particular by chronic disease. Rather than stories about disease, such reports are better characterized as a life interrupted by disease [57].

Even when an explanation can be represented in terms of possible shortcomings on the part of the family, or of specific individual characteristics and the personality of the children, this does not seem to be able to be questioned further. Once the disease is triggered, the complexity of the personal story fades to make room for a new narrative and inevitably also a new identity, the one associated with the anorexia disorder and pathological identity [47].

The personal implications, the contingent relationships within the family context, and the shared meanings that co-construct and cross it do not seem in any way contemplable. The emotions that emerge are those associated with anguish and a sense of helplessness in the face of something that is difficult, sometimes almost impossible, to contain and foresee.

A similar picture also emerges in a study by [10] on mothers’ home care experiences for adolescents with anorexia nervosa. In the interviews, the mothers speak of a life totally interrupted by the arrival of the disease and of a strong sense of isolation that invades them. Tensions emerge within the family unit as the attention on the part of the parent, or both, converges totally to the resolution of the problem and everything else remains in the background.

Consistent with the negative polarity of the first DS (models of problem explanation), the opposite polarity emerges as empty content: a void of meaning that is perfectly in line with the impossibility of providing further explanations to a story that seems to have stopped in the face of the emotional avalanche that the disorder has brought into the lives of these families.

The emotional burden of the experience of living and caring for a person with anorexia nervosa has been extensively documented in the literature [11,13,14,15,16]. In the study by Bezance and Holliday [10], the results also suggest that parents’ perceptions of their child and their future are strongly linked to their tendency to be too involved. In the studies by Whitney and Eisler [58], it is emphasized how family life can be reorganized around the eating disorder, with food dominating family interactions, activities, and relationships. As a result, families can begin to adapt to the behaviors of AN. In a previous study by the authors (2020) [59], it emerged that parents who became fundamentally immersed in the problem and totally committed to the reduction of symptoms were not able to perceive the role of the family context in the dynamics of the problem and in its recursive maintenance. The narratives that dominated were connected to the disease, its symptoms and its pathological identity.

If it is true, as Kirmayer argues, that identity can be fragmented through ruptures in the narrative, then we can also embrace the idea that the narration of these family members is consistent with that unique way of telling the self that emerged with respect to the onset of the disease. Pathological identity is the identity of the daughter as well as of the family that is fully embedded in it, a completely disrupted relationship between “internal and external reality” [57]. As pointed out by Salvatore and Venuleo [60], being captured by an emotion means representing the world in an absolutizing, homogenizing, and generalizing way. In this sense, it is possible to think that the experience of one’s child’s illness in these parents becomes a saturated space of thought not otherwise interpretable.

As regards the second DS (attribution of responsibility), two opposite ways of interpreting the attribution of responsibility emerge. The narratives of family members on this dimension seem to be interpretable and summarized in terms of “a context that cures and a society that makes you ill”.

On the one hand, the polarity of care services puts in the foreground the healthcare contexts, the professionals, and the healthcare system as a whole, recognized within the lens of its social mandate, namely that of treating and solving the problem. Once they are able to access the care services, the management of the disease seems completely delegated to the latter. The feelings of helplessness and incoherence found in the previous dimension seem to appear overwhelmingly with respect to the perception of one’s role in the treatment. According to Mattingly and Garro [57], individuals acquire narrative skills and the ways in which they are culturally embedded. In this case, the speeches of the family members seem to be completely imbued with medical codes. The studies of cultural psychology and medical anthropology have placed emphasis on how the analysis of the narrative can be a way to give us a certain vision of the world and to understand how this process of personal and cultural construction of the disease leads to a gradual stabilization, reification, and social incorporation of the same [57,61,62].

The attribution of responsibility in this sense therefore also emerges as a dimension of delegation towards professionals assigned to the definitive resolution of the disease.

In this sense, the results of a qualitative study [63] on the different descriptions used by doctors, nurses, and social professions to describe the family relationships of their patients are not surprising. Compared to the interpretative codes used, the results show that all three categories of professionals tend to use the terms “children,” “parents,” and “mothers” as descriptors of the family much less so than “fathers” and “couples.” As pointed out by the author himself, these forms of narrative mainly tend to catalyze the attention on parenthood rather than on the relationships that nourish the family context. Furthermore, doctors and nurses describe the tendency of parents to delegate the care of their children as a major concern in their work routine. In particular, it is nurses who emphasize the tendency of parents to delegate responsibilities, especially with the father, who is often absent or leaves the family in difficult times.

Delegation is defined as part of a wider family cultural deprivation or the result of internal problems of the couple (such as separation), or simply the result of the parents’ fears of the diagnosis being made explicit. On the other hand, social workers seem to be more attentive to emotional and psychological aspects and less to compliance, which remains equally important in the care process.

In the opposite polarity of the second DS (attribution of responsibility), the preponderant role of society, or of the outside world more generally, emerges, understood as the main culprits of one’s child’s illness.

This is a reference to the outside world being guilty of making people sick—on the one hand, a society that frames adolescents within perfect and homologating models of beauty, and on the other hand, the reference to the health emergency and to the conditions in which the children have been forced to live. Also noteworthy in this polarity is the absence of any reference to the family.

The specificity results support the above results by adding the differentiation between mothers and fathers. On the one hand, the specificities denote mothers completely engaged in the management phase of the disease, from the search for structures to the right treatment to be followed, and on the other hand, fathers oriented mainly on a pathologizing narrative of the disorder and on dimensions of delegation or a shift of responsibility towards the external, even with respect to one’s own family unit, such as the society, COVID-19, and quarantine.

Some studies show that the relationships between parents and children with anorexia inevitably undergo a change: classical studies have shown that female patients with EDs often reveal paternal emotional disengagement [64] characterized by reduced paternal expressiveness, withdrawal, or avoidance of conflict; mothers, on the other hand, tend to have a more empathic and understanding behavior, despite suffering a lot from the complex situation [65].

## 7. Conclusions

The results show that primary caregivers’ narratives construct anorexia nervosa as a sudden and disruptive disorder capable of upsetting the biographies of all family members.

Once the problem occurs, it is impossible to contain it. The only viable answer appears to be delegation to care services or the social context. On the one hand, services as a device capable of solving by themselves the set of problems that the AN has generated. The meanings of what it may represent and what may have originated and maintained over time are set aside. The attention is focused on the disorder, its symptomatic manifestations and the risks connected to the survival of the moment. In this frame, health professionals must have an omniscient power capable of intercepting the problem and solving it tout court. On the other hand, the gaze is turned to the danger inherent in the social context, seen as deceptive and guilty of spreading false ideals. In any case, primary caregivers seem to go through the need to shift attention away from the family context, its dynamics, and the contingency of meanings that run through it.

Other narratives do not seem to be thinkable. The discourses are repeated following a plot that appears already given: anorexia is an intractable problem, and only the intervention of professionals can help to contain it. Even the solution to the problem seems impossible because the threat remains constant. Consequently, the impossibility of preventing it renders the family unarmed concerning possible future relapses.

However, the research shows that the person and the family unit are essential in laying the foundations for an effective start, continuation, and conclusion of therapy [8]. The results of Bobbo’s study [63] also support the idea that it is not enough just to think about the multidisciplinary team, but it is essential to make all the actors involved in the treatment communicate with each other: professionals, family members, and patients.

Although family functioning was one of the first suggested factors relevant to eating disorders, Waller & Calam [66] point out that this is not the only possible conclusion. The family is also a means of transmitting pathological socio-cultural values to the potential patient and therefore does not necessarily represent a direct effect on the genesis of the problem.

However, to date, the family nucleus that surrounds the person involved in these disorders has only been marginally incorporated into the treatment path. Especially in the Italian context, the treatments are mainly parent training that takes place in the clinics that follow the AN children of these parents. It could be useful, for example, especially for primary caregivers, to promote paths that are not only attentive to the informative aspects of the disease. This often happens in psychoeducational groups [67] although its importance must be recognized. However, it is equally important to try to work with family members for the recognition of their personal, emotional, and relational resources.

As also pointed out by Mattingly and Garro [57] “at a pragmatic level, hearing narrative accounts is a principal means through which cultural understandings about illness—including possible causes, appropriate social responses, healing strategies, and characteristics of therapeutic alternatives—are acquired, confirmed, refined, or modified. […] Stories help to maintain narrative frameworks as a cultural resource for understanding illness experience” (p. 38).

## 8. Strengths and Limitations

It is necessary to recognize some limitations of this study. First, our sample is not representative; thus, the results cannot be generalized and must be correlated to a specific cultural context. In particular, the number of fathers is minimal, and any conclusion about them is limited. Nonetheless, the data on their poor compliance is essential to consider. Moreover, the nature itself of narratives does not allow us to extend the meaning specific to their narratives of health and disease.

Future studies could use the results to further confirm with the people involved. This would be beneficial both in terms of the reliability of the results and from the perspective of the care professionals and the participants. Both could benefit from sharing the narratives.

A further limitation is the absence of the narratives of the daughters and other family members involved, which, according to the study’s perspective, could provide additional clues on how the disorder is constructed and nurtured within the specific cultural context of belonging. It could be a further development of research on this issue.

Despite the limitations, our study deserves attention on a theoretical plan and practical level. In the theoretical plan, the study highlights how the analysis of people’s sensemaking may offer insight into the reasons guiding their ways of coping and reacting to disease challenges.

On a practical level, it provides suggestions to health professionals regarding the importance that narratives can have in construction compliance with patients and family members. Recognizing the narratives means recognizing not only the patient’s complexity but also the specific resources you can put in place to facilitate the results of the intervention.

The narratives of how the problem is experienced, what the treatment represents, what goals to achieve in the treatment, and what needs are present are not taken for granted. As Gergen points out [68], narratives are important not because they provide an accurate picture of what really happened, but because they show how people understand and live their experiences and because they are ways to participate in the practice of a given culture actively. As Eisenberg suggested [69], the decision to seek medical advice is itself a request for interpretation, emphasizing how co-constructing the discourse between the patient and the healer is an important part of clinical care and psychiatric practice: “Patient and doctor together reconstruct the meanings of events in a shared mythopoesis. […] Once things fall into place, once experience and interpretation appear to coincide, once the patient has a coherent “explanation”, which leaves him no longer feeling the victim of the inexplicable and the uncontrollable, the symptoms are usually exorcised” (p. 245).

## Figures and Tables

**Table 1 healthcare-10-01353-t001:** Socio-demographic characteristics of the respondents, disaggregated for role.

		*Role*				
*Variables*		Mothers	Fathers	Total	Chisquare	*p*-Value
Marital status	Married	(12) 63.2%	1 (5.3%)	13 (68.4%)	6.041	0.049
	Divorced	(4) 21.1%	1 (5.3%)	5 (26.3%)		
	Widowed	(0)	1 (5.3%)	1 (5.3%)		
Level of schooling	Lower primary school	1 (5.3%)	(0)	1 (5.3%)	0.270	0.966
	Lower secondary school	(4) 21.1%	1 (5.3%)	5 (26.3%)		
	Secondary school	(6) 31.6%	1 (5.3%)	7 (36.8%)		
	University degree	5 (26.3%)	1 (5.3%)	(6) 31.6%		
Employment	Employed	(6) 31.6%	(0)	(6) 31.6%	8.972	0.072
	Freelance professionals	(4) 21.1%	2 (10.5)	(6) 31.6%		
	Unemployed	3 (15.8%)	(0)	3 (15.8%)		
	Housewives	3 (15.8%)	(0)	3 (15.8%)		
	Retired	(0)	1 (5.3%)	1 (5.3%)		

**Table 2 healthcare-10-01353-t002:** Characteristics of the dataset.

	No.
Texts in the corpus	106
Elementary contexts (EC)	263
Types	2253
Lemmas	1845
Occurrences (Tokens)	10,447
Threshold of lemma selection	4
Lemmas in analysis	180 (208)

Notes—Texts in the corpus: number of answers to the open question (corresponding to the number of participants) inserted in the text analysis; Elementary context: sections of text (e.g., sentences, paragraphs, or short texts) characterized by the same keyword patterns; Types: total number of words (i.e., including all linguistic forms) contained in the general corpus; Lemmas: words transformed into headword; Occurrences (Tokens): frequencies of a single lexical unit; Threshold of lemma selection: the value selected to include the lemma in the analysis; Lemmas in analysis: number of headwords inserted in analysis.

**Table 3 healthcare-10-01353-t003:** First Factorial Dimension—DS.

Models of Problem Explanation			
Onset (−)		(no explanation) (+)	
Lemmas	Test value *	Lemmas	Test value *
Lack	−18.7882	-	-
Familiar	−14.3825		
Affection	−13.9494		
Self-esteem	−8.9159		
Problematic	−4.4027		
To believe	−3.3084		
Be born	−2.8362		
Question	−2.7836		
Pathology	−2.7501		
To prevent	−2.5219		
Need to	−2.5176		
Greater	−2.2645		
Family	−2.0764		

* Highest levels of association standard scores (V-Test).

**Table 4 healthcare-10-01353-t004:** Second Factorial Dimension—DS.

Attribution of Responsibility			
health care services (−)		Society (+)	
Lemmas	Test value *	Lemmas	Test value *
Doctor	−6.4948	Go out	4.5532
Hospitalize	−5.4954	Food	4.5418
Polyclinic	−5.4435	Love	4.3694
Private	−5.328	World	3.9998
Salerno	−4.6301	Today	3.896
To accept	−4.5247	Society	3.4907
Send	−4.3906	To remain	3.4878
Asl	−4.3752	Hunger	3.4743
Antimo	−4.3186	Psychological	3.1393
Structure	−4.2642	Fear	3.1109
Call	−4.2613	Figure	3.0639
Psychologist	−4.2379	Model	3.0185
To address	−4.1458	To be able to	3.0036
Know	−3.952	Image	2.9712
Solve	−3.8744	Brain	2.814
I disturb	−3.8301	Body	2.7622
Center	−3.7027	To think	2.7202
Known	−3.6844	Son	2.6433
Week	−3.6781	Husband	2.5407
Food	−3.2595	Become	2.4018

* Highest levels of association standard scores (V-Test).

**Table 5 healthcare-10-01353-t005:** Specificity in mothers and fathers.

	SPEC (+)					SPEC (−)				
Categorial Variable	LEMMA *	SUB	TOT	CHI2	(*p*)	LEMMA	SUB	TOT	CHI2	(*p*)
**Mother**	Doctor	29	30	6.54	0.01	To live	12	31	−19.04	0.00
	-	-	-	-	-	Brain	0	5	−16.99	0.00
	-	-	-	-	-	Disease	4	12	−13.21	0.00
	-	-	-	-	-	Sensation	1	5	−9.33	0.00
	-	-	-	-	-	Good	1	5	−9.33	0.00
	-	-	-	-	-	Social	1	5	−9.33	0.00
	-	-	-	-	-	Greater	1	5	−9.33	0.00
	-	-	-	-	-	Experience	12	23	−8.29	0.00
	-	-	-	-	-	Family	7	15	−8.02	0.00
	-	-	-	-	-	Hold	3	8	−7.20	0.00
**Father**	To live	16	31	19.04	0.00	Doctor	1	30	−6.54	0.01
	Disease	8	12	13.21	0.00	To follow	0	18	−5.35	0.02
	Sensation	4	5	9.33	0.00	To be able to	0	17	−5.05	0.02
	Good	4	5	9.33	0.00	Nutritionist	0	13	−3.86	0.05
	Greater	4	5	9.33	0.00	-	-	-	-	-
	Social	4	5	9.33	0.00	-	-	-	-	-
	Experience	11	23	8.30	0.00	-	-	-	-	-
	Family	8	15	8.02	0.00	-	-	-	-	-
	Hold	5	8	7.20	0.00	-	-	-	-	-
	Sense	4	6	6.59	0.01	-	-	-	-	-

* LEMMA = specific lexical units (over/under used); SUB = occurrences of each LEMMA in the subset; TOT = occurrences of each LEMMA in the corpus or in the two compared subsets (see 2.1 below); CHI2 = CHI square value (or VTEST = Test Value); (*p*) = probability associated with the chi square value (def = 1).

**Table 6 healthcare-10-01353-t006:** Exclusive Words.

Mather		Father	
Lemma	Occurrence	Lemma	Occurrence
Own	29	Brain	5
To follow	18	Isolated	4
To be able to	17	Take advantage	3
Nutritionist	13	Depression	3
Ugly	12	Wife	3
To start	12	Forty	3
To accept	10	Put back	3
Face up to	10	You	3
Hard	10	Insiders	2
Physicist	9	Analyses	2
Lose	9	To surround	2
Polyclinic	9	Contribute	2
Personal	8	Domestic partnership	2
Need	7	Depressive	2
Character	7	Disorientation	2

## Data Availability

The data that support the findings of this study are available on request from the corresponding author. The data are not publicly available due to privacy or ethical restrictions.

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
