# Peer review of "Caring for Daughters with Anorexia Nervosa: A Qualitative Study on Parents’ Representation of the Problem and Management of the Disorder"

_healthcare, 2022, doi:10.3390/healthcare10071353_

Round 1
Reviewer 1 Report
This is a very interesting study which explores the implicit theories by which PC of patients diagnosed with AN understand the eating disorder and interpret their role in treating and managing the problem.
Authors employed in-depth, semi-structured, and open interview questions to 19 caregivers. Data was interpreted using LCA.
They extracted two dimensions from the analyses: 1) models of explanation of the problem and 2) perspectives on the attribution of responsibility.
The analyses show the difficulties of PC in exploring the emotional dynamics of the problem and the tendency to take out of the family context every possible representation of the role that it can play in the maintenance and evolution of the disorder.
However, I found some issues to be addressed before acceptance. See my comments divided by section.
Abstract OK
Intro OK
Methods
Positive and negative polarities (meaning) which emerging from the LCA analyses should be explained in Methods’ section. I found only vague mention of the topic in p5L197. Both polarities are very developed into Results’ section, so it could be a confounding issue for non-specialized readers in LCA.
Results
Because of the reduced numbers of fathers, comparisons with mothers in Tables 5 should be treated with caution or directly removed. On the other hand, Table 6 is OK from a qualitative point of view.
Discussion
Results are well discussed. However, staring from a personal and esthetical point of view parent phrases should be removed from the discussion and conclusions. This is not mandatory, but it isn’t elegant either. Those sentences should be placed in the Results’ section.
A subhead analyzing strengths and limitations of the study should be added.
Finally, conclusions should summarize the findings, so I encourage authors to do this. I see this section as an extension of the discussion.
Author Response
REVIEWER #1’ S COMMENT-1
Intro OK
Methods
Positive and negative polarities (meaning) which emerging from the LCA analyses should be explained in Methods’ section. I found only vague mention of the topic in p5L197. Both polarities are very developed into Results’ section, so it could be a confounding issue for non-specialized readers in LCA.
AUTHORS’ ANSWER
The method section has been further enriched, and the LCA procedure has been detailed.
ACASM procedure consists of the following steps:
First, through an automated procedure (see, Lancia, 2012), the textual corpus of the narratives is segmented into Elementary Context Units (ECUs). Second, lexical forms (i.e., any string of characters comprised between two blank spaces) present in the ECUs are subject to a lemmatization procedure (i.e., lexical forms “son,” “sons” and “daughter” are classified as the lemma “child”). Lexical forms devoid of meaning (e.g., or, so, because, "that is" etc.) are automatically excluded. A digital matrix of the text is generated, having as rows the ECU and the lemmas as columns. The ij-th cell assumes a value of 1 if the j-th lemma is contained in the i-th ECU; otherwise, a value of 0 is assigned.
LCA is a factor analysis procedure that works on nominal data (Benzécri, 1973). In general terms, the method breaks down the whole lexical variability (i.e. the distribution of the lemmas presented in the corpus) in terms of a multidimensional structure of op-posed factorial polarities.
REVIEWER #1’ S COMMENT-2
Results
Because of the reduced numbers of fathers, comparisons with mothers in Tables 5 should be treated with caution or directly removed. On the other hand, Table 6 is OK from a qualitative point of view.
AUTHORS’ ANSWER
A sentence was added in the discussion of the results, and in the limits section, the issue was further addressed.
Discussion
Although these data should be treated with extreme caution, given the small number of respondents, on the other hand…
Limitations
In particular, the number of fathers is minimal, and any conclusion about them is not convenient. Nonetheless, the data on their poor compliance is essential to consider. Moreover, the nature itself of narratives does not allow us to extend the meaning specific to their narratives of health and disease.
REVIEWER #1’ S COMMENT-3a
Discussion
Results are well discussed. However, staring from a personal and esthetical point of view parent phrases should be removed from the discussion and conclusions. This is not mandatory, but it isn’t elegant either. Those sentences should be placed in the Results’ section.
AUTHORS’ ANSWER
Thanks for the suggestion. We have removed the sentences and moved them into the results section.
REVIEWER #1’ S COMMENT-4-5
A subhead analyzing strengths and limitations of the study should be added.
Finally, conclusions should summarize the findings, so I encourage authors to do this. I see this section as an extension of the discussion.
Reviewer 2 Report
Dear Editors,
Thank you for the opportunity to review the manuscript, Caring for daughters with anorexia nervosa: a qualitative study on parents' representation of the problem and management of the disorder, for consideration of publication in Healthcare. Authors present and original qualitative study of the narratives of 19 parents of daughters being treated for AN. They explored how these parents represented and managed their child’s disease – to identify the narratives of parents. Findings suggested two factors with differing narratives by mothers and fathers. Findings were supported by previous research findings. I recommend the manuscript be further considered for publication after revision: - specifically,
(1) There does not seem to be a need for (dis) in the abstract.
(2) Consider revising to use person first language (e.g., instead of Eds member say member with EDs).
(3) Page 1, states that families with members with EDs have worse function and then reports reported perceptions. Who is the respondent reporting these concerns – parents, siblings? Identifying the age of the population of interest to this study and literature used as the foundation of this study. Initially ‘member’ is used and later ‘children’ and ‘adolescent’ are used. Later on page 2 line 79 the term shifts to ‘patients”
(4) Page 2, The statement on line 49 needs some clarity. “Caregivers,…, reduce the use of adaptive coping strategies” by whom – their own or the person with AN?
(5) Authors refer to the person with AN as ‘daughter’ it would be helpful to establish the prevalence of AN for males and females before using the term daughter.
(6) Throughout this section, authors report findings of previous studies. It would be helpful for the reader to understand more about the studies’ contexts – how many parents contributed to this general finding, as an example. Often these studies may have a small sample, which can be problematic to support these types of statements of generalizability. Likewise, consider tempering language when small sample studies are being referenced – for example, page 2 line 75 says “mothers often” consider “mothers may”.
(7) Authors use the term ‘dysfunctional parents’ – it is unclear if the statement on line 67 is describing what is meant by the term or attributing characteristics. Revision is needed for clarity – a definition of dysfunctional parents. Also, the term previously used was caregiver. It there a difference between the use of caregiver and parent?
(8) The literature review seems to go back and forth between AN and ED. Connections for when the terms shift would be helpful as AN is only one type of ED.
(9) Line 82 begins ‘given the lack of in-depth studies’ – this follows the review of the studies without pointing out limitations as to the depth or a mention of the dearth of research on this topic. So – it isn’t given, as it has not been established in the introduction to that point.
(10) Define ‘narratives’ under theoretical framework. Authors indicate how they are used but no description for the meaning is applied.
(11) Cite the theorists for the cultural and social constructive perspectives applied.
(12) Remove the colloquialism “it goes without saying” line 94.
(13) The theoretical framework would benefit from editing to reduce the length of sentences to convey clear thoughts and connections. For example, the paragraph stating line 119 is 10 lines and only 2 sentences with multiple clauses. The paragraph below it is only one sentence – 6 lines. The theoretical frame is critical to the study and therefore should be very clearly articulated and connected to the intent of the study.
(14) Participants – authors state the participants were being treated in an outpatient clinic, however they then stated they were hospitalized. It is that the patients access outpatient after release from hospital. If so, please clarity.
(15) How did the sample size go from 26 parents (mother and father of 13 patients) to 19 parents as the final sample? Did some not agree to participate? Dropped out? It looks like fathers more likely did not participate. Some background on why that might be would be relevant to the background information. Authors touch on differences but are fathers less likely to participate in the child’s care -which could account for the lower participation by fathers. This is also relevant on terms of reported evidence about line 475 about fathers often leaving the family.
(16) Given the gross difference in participation between mothers and fathers. Consider having 2 columns for table one reporting each group separately.
(17) The positionality of the researchers engaged in analysis is needed. Further a description of who engaged in the analysis is needed.
(18) Authors describe their coding procedures well, however, there is a need to address issues of trustworthiness of the data – were findings confirmed with participants, were other data used to triangulate findings. If not this is a limitation that should be addressed in the discussion.
(19) What was the consensus procedure used to resolve coding conflicts?
(20) For the first factorial, some description of ‘prevention’ would be helpful as it is labeled as problematic. However, typically prevention would be desired. This is also a direct question in the interview – so that it emerged is not surprising. What is to be made of this?
(21) I appreciated the results disaggregated/analyzed by parent (mothers and fathers) given the points made in the introduction – a justification for why this was not dome for factor 1 is needed.
(22) The discussion focused on a shared narrative for these families (e.g., a life disrupted). Were there examples of counternarratives expressed by any of the participants?
(23) There is discussion of cultural implications on the narrative – did all participants represent a similar cultural group? A description under participants would be helpful to apply to the interpretation of the data.
(24) We learn that the study was conducted during COVID just before the conclusion. It seems relevant to provide that context in the introduction and report any evidence of increased cased of EDs or AN for adolescents (and girls in this case).
(25) Limitations and implications of this work are needed. All studies have limitations that help further the field of study. Identifying them is an important part of the research process. Further, how might these findings be useful for working with families or designing treatment? What are the next steps in this line of inquiry. Understanding the narratives seems to be an early step in the processes of learning about how to better serve this population.
Respectfully,
Author Response
REVIEWER #2’ S COMMENT 25
Limitations and implications of this work are needed. All studies have limitations that help further the field of study. Identifying them is an important part of the research process. Further, how might these findings be useful for working with families or designing treatment? What are the next steps in this line of inquiry. Understanding the narratives seems to be an early step in the processes of learning about how to better serve this population.
AUTHORS’ ANSWER
The conclusions were revised based on the reviewers' comments.
The limitations, strengths analysis and implications of this work have been added at the end of the document.
Conclusion
The results show that primary caregivers' narratives construct anorexia nervosa as a sudden and disruptive disorder capable of upsetting the biographies of all family members.
Once the problem occurs, it is impossible to contain it. The only viable answer appears to be delegation to care services or the social context. On the one hand, services as a device capable of solving by themselves the set of problems that the AN has generated. The meanings of what it may represent and what may have originated and maintained over time are set aside. The attention is focused on the disorder, its symptomatic manifestations and the risks connected to the survival of the moment. In this frame, health professionals must have an omniscient power capable of intercepting the problem and solving it tout court. On the other hand, the gaze is turned to the danger inherent in the social context, seen as deceptive and guilty of spreading false ideals. In any case, primary caregivers seem to go through the need to shift attention away from the family context, its dynamics, and the contingency of meanings that run through it.
Other narratives do not seem to be thinkable. The discourses are repeated following a plot that appears already given: anorexia is an intractable problem, and only the intervention of professionals can help to contain it. Even the solution to the problem seems impossible because the threat remains constant. Consequently, the impossibility of preventing it renders the family unarmed concerning possible future relapses.
However, the research shows the person, and the family unit are essential to lay the foundations for an effective start, continuation, and conclusion of therapy [51]. The results of Bobbo's study [48] also support the idea that it is not enough just to think about the multidisciplinary team, but it is essential to make all the actors involved in the treatment communicate with each other: professionals, family members, and patients.
Although family functioning was one of the first suggested factors relevant to eating disorders (e.g., Lasegue, 1873), Waller & Calam point out that this is not the only possible conclusion. The family is also a means of transmitting pathological socio-cultural values ​​to the potential patient and therefore does not necessarily represent a direct effect on the genesis of the problem.
However, to date, the family nucleus that surrounding the person involved in these disorders has only been marginally incorporated into the treatment path. Especially in the Italian context, the treatments are mainly parent training that takes place in the clinics that follow the children with AN of these parents. It could be useful, for example, especially for primary caregivers, to promote paths that are not only attentive to the informative aspects of the disease. This often happens in psychoeducational groups [54] although its importance must be recognized. However , it is equally important to try to work with family members for the recognition of their personal, emotional, and relational resources.
As also pointed out by Mattingly and Garro, "at a pragmatic level, hearing narrative accounts is a principal means through which cultural understandings about illness — including possible causes, appropriate social responses, healing strategies, and characteristics of therapeutic alternatives—are acquired, confirmed, refined, or modified. [...] Stories help to maintain narrative frameworks as a cultural resource for understanding illness experience" (p.38)
- Strengths and limitations
It is necessary to recognize some limitations of this study. First, our sample is not representative; thus, the results cannot be generalized and must be correlated to a specific cultural context. In particular, the number of fathers is minimal, and any conclusion about them is limited. Nonetheless, the data on their poor compliance is essential to consider. Moreover, the nature itself of narratives does not allow us to extend the meaning specific to their narratives of health and disease.
Future studies could use the results to further confirm with the people involved. This would be beneficial both in terms of the reliability of the results and from the perspective of the care professionals and the participants. Both could benefit from sharing the narratives.
A further limitation is the absence of the narratives of the daughters and other family members involved, which according to the study's perspective, could provide additional clues on how the disorder is constructed and nurtured within the specific cultural context of belonging. It could be a further development of research on this issue.
Despite the limitations, our study deserves attention on a theoretical plan and practical level. In the theoretical plan, the study highlights how the analysis of people's sensemaking may offer insight into the reasons guiding their ways of coping and reacting to disease challenges.
On a practical level, it provides suggestions to health professionals regarding the importance that narratives can have in construction compliance with patients and family members. Recognizing the narratives means recognizing not only the patient's complexity but also the specific resources you can put in place to facilitate the results of the intervention.
The narratives of how the problem is experienced, what the treatment represents, what goals to achieve in the treatment, and what needs are present are not taken for granted. As Gergen points out [52], narratives are important not because they provide an accurate picture of what really happened but because they show how people understand and live their experiences and because they are ways to participate in the practice of a given culture actively. As Eisenberg suggested [53], the decision to seek medical advice is itself a request for interpretation, emphasizing how co-constructing the discourse between the patient and the healer is an important part of clinical care and psychiatric practice: "Patient and doctor together reconstruct the meanings of events in a shared mythopoesis. […] Once things fall into place, once experience and interpretation appear to coincide, once the patient has a coherent "explanation", which leaves him no longer feeling the victim of the inexplicable and the uncontrollable, the symptoms are usually exorcised" (p. 245).
REVIEWER #2’ S COMMENT-1
(1) There does not seem to be a need for (dis) in the abstract.
AUTHORS’ ANSWER
(dis) was removed from the abstract
REVIEWER #2’ S COMMENT-2
Consider revising to use person first language (e.g., instead of Eds member say member with Eds).
AUTHORS’ ANSWER
We have modified the term in the body of the paper.
REVIEWER #2’ S COMMENT-3
Page 1, states that families with members with EDs have worse function and then reports reported perceptions. Who is the respondent reporting these concerns – parents, siblings? Identifying the age of the population of interest to this study and literature used as the foundation of this study. Initially ‘member’ is used and later ‘children’ and ‘adolescent’ are used. Later on page 2 line 79 the term shifts to ‘patients”
AUTHORS’ ANSWER
We have included additional data found in the scientific literature to specify who was involved in the research.
“Generally, parents of child with an EDs member report worse family functioning and specific problems with communication, flexibility, and cohesion [6,7]. In a study [8] in which the opinions of parents and siblings of AN patients (n°34 female patients; mean age 15.7) were evaluated, it emerged that parents rated their families with a high level of conflict and with a more dysfunctional general functioning especially with regard to the relationship with the mother.”
REVIEWER #2’ S COMMENT-4
Page 2, The statement on line 49 needs some clarity. “Caregivers,…, reduce the use of adaptive coping strategies” by whom – their own or the person with AN?
AUTHORS’ ANSWER
We have included the specification:
“Caregivers, according to Coomber and King [11], reduce the use of adaptive coping strategies and implement moderate levels of maladaptive coping strategies, such as self-blame, denial, and behavioral disengagement with their cared-for child.”
REVIEWER #2’ S COMMENT-5
Authors refer to the person with AN as ‘daughter’ it would be helpful to establish the prevalence of AN for males and females before using the term daughter.
AUTHORS’ ANSWER
We reported data on the incidence and prevalence of AN among males and females. Since there is a higher percentage in the female gender and that all patients involved are female, we clarified the use of the term "daughter" in the family context.
“In January 2022, the Italian National Institute of Health showed data from the first census regarding the prevalence of AN in Italy [4]. According to the data, with respect to the most frequent diagnoses AN is represented in 42.3% of cases, Bulimia Nervosa in 18.2% and Binge Eating Disorder in 14.6%. AN is 90% present in the female gender 90% compared to 10% of males; the reference age range is 59% between 13 and 25 years and 6% <12 years.
Additionally, the COVID-19 pandemic has negatively impacted mental health prob-lems; in particular, there was a 30% increase in cases of AN [5].”
REVIEWER #2’ S COMMENT-6
Throughout this section, authors report findings of previous studies. It would be helpful for the reader to understand more about the studies’ contexts – how many parents contributed to this general finding, as an example. Often these studies may have a small sample, which can be problematic to support these types of statements of generalizability. Likewise, consider tempering language when small sample studies are being referenced – for example, page 2 line 75 says “mothers often” consider “mothers may”.
AUTHORS’ ANSWER
We have included sample size and mean age of the studies noted in the scientific literature.
“Parks et al. [15] conducted a comparative study to assess caregiving experience and coping strategies among groups of parents caring for their children with different diagnoses. The study sample consisted of 48 mothers (mean age 44.88) and 44 fathers (mean age 47.45) of 50 patients diagnosed with EDs (98% daughters; mean age 14.78); 46 mothers (mean age 49.95) and 36 fathers (mean age 51.06) of 46 patients diagnosed with substance abuse (82.6% sons; mean age 18.3) and 63 mothers (mean age 47.56) and 50 fathers (mean age 49.72) of 66 healthy adolescents (all daughters; mean age 14.39). Analysis of the results showed that mothers of adolescents with EDs adopt self-sustaining strategies that focus on problems more than mothers of healthy ado-lescents, and this can lead to hostility and over-involvement. Some differences in pa-rental reactions between mothers and fathers have been found, the former being more emotionally involved, less critical and more prone to seeking supportive social rela-tionships than fathers.
Differences between mothers and fathers are also highlighted between negative emotions and a sense of self-efficacy: mothers often manifest fear, and this produces accommodating attitudes towards the disease of their daughter AN and predicts their sense of self-efficacy; among fathers, neither fear nor self-efficacy predicts disease [16].”
REVIEWER #2’ S COMMENT-7
Authors use the term ‘dysfunctional parents’ – it is unclear if the statement on line 67 is describing what is meant by the term or attributing characteristics. Revision is needed for clarity – a definition of dysfunctional parents. Also, the term previously used was caregiver. It there a difference between the use of caregiver and parent?
AUTHORS’ ANSWER
We clarified the sentence as the term dysfunctional is to refer to maladaptive coping strategies.
REVIEWER #2’ S COMMENT-8
The literature review seems to go back and forth between AN and ED. Connections for when the terms shift would be helpful as AN is only one type of ED.
AUTHORS’ ANSWER
We clarified where it was correct to use the term EDs (for studies that generalized the data) and where the term AN for specific studies.
REVIEWER #2’ S COMMENT-9
Line 82 begins ‘given the lack of in-depth studies’ – this follows the review of the studies without pointing out limitations as to the depth or a mention of the dearth of research on this topic. So – it isn’t given, as it has not been established in the introduction to that point.
AUTHORS’ ANSWER
Thank you for the valuable suggestion. We decided to remove the sentence as it was not clarifying the exploratory nature of our study.
REVIEWER #2’ S COMMENT-10
Define ‘narratives’ under theoretical framework. Authors indicate how they are used but no description for the meaning is applied.
AUTHORS’ ANSWER
The term “narratives” was further defined, and a reference literature was added.
According to Polkinghorne narrative is the "linguistic form uniquely suited for dis-playing human existence as situated action. Narrative descriptions exhibit human activity as purposeful engagement in the world." (p.5) Frank notes that while people tell their own stories of disease, they compose these stories by adopting and combining the narrative types that cultures make available to them. In this sense, narrative tellings are not mere conversational realities; but them-selves represent constituents of ongoing and often institutionalized patterns of social conduct. The narratives of health and illness, are both personal and social, are how people make sense of their experiences and are part of general social relationships and cultural values that characterize a specific life context.
REVIEWER #2’ S COMMENT-11
Cite the theorists for the cultural and social constructive perspectives applied.
AUTHORS’ ANSWER
The theorists were added
Bruner, J. (1990). Acts of meaning. Harvard university press.
Gergen, K. J. (1992). The social constructionist movement in modern psychology. American Psychologist, 40 (3), 266-275.
Harré, R. (1986). The step to social constructionism. Children of social worlds, 287-96.
Salvatore, S., Freda, M. F., Ligorio, B., Iannaccone, A., Rubino, F., Scotto di Carlo, M., et al. (2003). Socioconstructivism and theory of the unconscious. A gaze over a research horizon. European Journal of School Psychology, 1(1), 9–36.
Valsiner, J. (2007). Culture in minds and societies: Foundations of cultural psychology. SAGE Publications India.
REVIEWER #2’ S COMMENT-12
Remove the colloquialism “it goes without saying” line 94.
AUTHORS’ ANSWER
The colloquialism “it goes without saying” was removed
REVIEWER #2’ S COMMENT-13
The theoretical framework would benefit from editing to reduce the length of sentences to convey clear thoughts and connections. For example, the paragraph stating line 119 is 10 lines and only 2 sentences with multiple clauses. The paragraph below it is only one sentence – 6 lines. The theoretical frame is critical to the study and therefore should be very clearly articulated and connected to the intent of the study.
AUTHORS’ ANSWER
Thank you for your suggestion. The sentences have been reduced
REVIEWER #2’ S COMMENT-14
Participants – authors state the participants were being treated in an outpatient clinic, however they then stated they were hospitalized. It is that the patients access outpatient after release from hospital. If so, please clarity.
AUTHORS’ ANSWER
All patients are from a previous hospital stay and have a diagnosis of AN, in accordance with DSM 5 criteria.
REVIEWER #2’ S COMMENT-15
How did the sample size go from 26 parents (mother and father of 13 patients) to 19 parents as the final sample? Did some not agree to participate? Dropped out? It looks like fathers more likely did not participate. Some background on why that might be would be relevant to the background information. Authors touch on differences but are fathers less likely to participate in the child’s care -which could account for the lower participation by fathers. This is also relevant on terms of reported evidence about line 475 about fathers often leaving the family.
AUTHORS’ ANSWER
The number of fathers who did not join the research was clarified, and the reason for their absence.
Of the 12 couples involved (a widowed father and a divorced mother unaccompanied by her daughter's father), only in 2 cases did the fathers decide to participate in the research. The remaining ten fathers agreed to participate but delegated it to the mother.
REVIEWER #2’ S COMMENT-16
Given the gross difference in participation between mothers and fathers. Consider having 2 columns for table one reporting each group separately.
AUTHORS’ ANSWER
As suggested, a role-disaggregated table was reconstructed
REVIEWER #2’ S COMMENT-17
The positionality of the researchers engaged in analysis is needed. Further a description of who engaged in the analysis is needed.
AUTHORS’ ANSWER
The positionality of the researchers engaged in the analysis was added.
In a double-blind procedure, the authors, two development and Education Psychologists and one clinical psychologist, all with previous experience in qualitative research, defined the labels and commented on the results.
REVIEWER #2’ S COMMENT-18
Authors describe their coding procedures well, however, there is a need to address issues of trustworthiness of the data – were findings confirmed with participants, were other data used to triangulate findings. If not this is a limitation that should be addressed in the discussion.
AUTHORS’ ANSWER
The coding procedure was further detailed and the issues of trustworthiness of the data were addressed in the limitations section.
REVIEWER #2’ S COMMENT-19
What was the consensus procedure used to resolve coding conflicts?
AUTHORS’ ANSWER
In a double-blind procedure, the authors, two development and Education Psychologists and one clinical psychologist, all with previous experience in qualitative research, defined the labels and commented on the results. After reading the analysis as a whole, all authors performed a second in-depth reading of the corpus of interviews, discussed and compared the results until an agreement was reached.
REVIEWER #2’ S COMMENT-20
For the first factorial, some description of ‘prevention’ would be helpful as it is labeled as problematic. However, typically prevention would be desired. This is also a direct question in the interview – so that it emerged is not surprising. What is to be made of this?
AUTHORS’ ANSWER
A description of ‘prevention’ with related examples has been added in the results section.
Even the discourse on prevention seems polarized on social or cultural gaps. Anorexia is represented as something that happens suddenly, challenging to contain before its manifestation, or the response to society's shortcomings, guilty of promoting exaggerated dictates of beauty and not providing the necessary information.
I don't know how to prevent it because maybe I found myself inside, catapulted out of no-where (Mother, 49 years old).
... so more information, less perfect projected images. So more helpful information to prevent this should be the foundation of our society (Mother, 48 years old).
REVIEWER #2’ S COMMENT-21
I appreciated the results disaggregated/analyzed by parent (mothers and fathers) given the points made in the introduction – a justification for why this was not dome for factor 1 is needed.
AUTHORS’ ANSWER
More details on the procedures have been added in the analysis section. The disaggregated results relate to the entire corpus of text.
Finally, with the support of the same software, a Specificity Analysis (SA) was applied to the entire corpus of text to identify and investigate other differences or similarities in the respondent's discourses.
REVIEWER #2’ S COMMENT-22
The discussion focused on a shared narrative for these families (e.g., a life disrupted). Were there examples of counternarratives expressed by any of the participants?
AUTHORS’ ANSWER
It is an interesting aspect of the discourses of family members we interviewed in this study. The narratives seemed to repeat themselves. The suggestion was added in the conclusions section.
Other narratives do not seem to be thinkable. The discourses are repeated following a plot that appears already given: anorexia is an intractable problem, and only the intervention of professionals can help to contain it. Even the solution to the problem seems impossible because the threat remains constant. Consequently, the impossibility of preventing it renders the family unarmed concerning possible future relapses.
REVIEWER #2’ S COMMENT-23
There is discussion of cultural implications on the narrative – did all participants represent a similar cultural group? A description under participants would be helpful to apply to the interpretation of the data.
AUTHORS’ ANSWER
Yes, they represent a similar cultural group. All participants came from different areas of the same region. The care service is a regional reference center.
REVIEWER #2’ S COMMENT-24
We learn that the study was conducted during COVID just before the conclusion. It seems relevant to provide that context in the introduction and report any evidence of increased cased of EDs or AN for adolescents (and girls in this case).
AUTHORS’ ANSWER
In introduction, evidence of cases in a period of pandemic crisis was reported.